# CO oxidation activity of non-reducible oxide-supported mass-selected few-atom Pt single-clusters

Atsushi Beniya [1✉], Shougo Higashi [1✉], Nobuko Ohba[1], Ryosuke Jinnouchi[1], Hirohito Hirata[2] & Yoshihide Watanabe[1]

Platinum nanocatalysts play critical roles in CO oxidation, an important catalytic conversion process. As the catalyst size decreases, the influence of the support material on catalysis increases which can alter the chemical states of Pt atoms in contact with the support. Herein, we demonstrate that under-coordinated Pt atoms at the edges of the first cluster layer are rendered cationic by direct contact with the $Al_2O_3$ support, which affects the overall CO oxidation activity. The ratio of neutral to cationic Pt atoms in the Pt nanocluster is strongly correlated with the CO oxidation activity, but no correlation exists with the total surface area of surface-exposed Pt atoms. The low oxygen affinity of cationic Pt atoms explains this counterintuitive result. Using this relationship and our modified bond-additivity method, which only requires the catalyst–support bond energy as input, we successfully predict the CO oxidation activities of various sized Pt clusters on $TiO_2$.

[1] Toyota Central R&D Labs, Inc., 41-1 Yokomichi, Nagakute, Aichi 480-1192, Japan. [2] Toyota Motor Corporation, 1200 Mishuku, Susono, Shizuoka 410-1193, Japan. ✉email: beniya@mosk.tytlabs.co.jp; shigashi@mosk.tytlabs.co.jp

Precious metal nanoclusters, with diameters of a few nanometres or less, are of particular importance as catalysts in heterogeneous catalytic conversions such as the oxygen reduction reaction for fuel cells[1,2], hydrogenation reactions in organic chemistry[3,4], and CO oxidation (Eq. (1)), which are designed to control emissions from combustion sources[5–7].

$$CO + 1/2O_2 \rightarrow CO_2 \tag{1}$$

Reducing the catalyst size increases the number of atoms that are exposed on their surface (Supplementary Fig. 1) and therefore can increase the catalytic efficiency in terms of mass loading. However, when the diameter of the catalytic cluster is in the range of a few nanometres, the interactions between the cluster and the support material holding the cluster become significant[8,9]. For a nanocluster with a diameter of ~1 nm that consists of ~20 atoms, roughly 70% of the atoms are in direct contact with the support material, and consequently, the effect of the support material on the catalytic reaction is not negligible[5]. To clarify this effect quantitatively, a combination of model systems, in which uniform mass-selected clusters are distributed on well-defined single crystal metal oxide support substrates, simple catalytic reactions such as CO oxidation, and surface-sensitive analytic techniques are required[6,7,10,11].

Two types of surface reaction mechanism have been discussed for CO oxidation over platinum group metal (PGM) nanocatalysts: the Mars–van Krevelen mechanism, in which the reaction involves lattice oxygen in the support, and the Langmuir–Hinshelwood (L-H) mechanism, in which the reaction occurs on the catalyst surface without the involvement of oxygen in the support[5,12–14]. The type of reaction that occurs depends on the reducibility of the support material. $Al_2O_3$ is a non-reducible oxide and previous studies have confirmed that L-H-type CO oxidation occurs on Pt nanoparticle (NP) catalysts with diameters of approximately 1–20 nm supported on $Al_2O_3$[13,14]. Although infrared spectroscopy has confirmed the existence of oxidized Pt atoms in the NPs[15], it is not clear whether the Pt atoms on the support are fully oxidized or partly oxidized or how the support affects the chemical states of Pt atoms in contact with the support and the resulting CO oxidation activity. An in-depth understanding of the oxidation state of Pt on such oxides and its relationship to the CO oxidation activity is needed.

Infrared reflection absorption spectroscopy (IRAS) is often used as a tool for identifying the chemical states of Pt nanoclusters[16–19]. If oxidized Pt exists in a nanocluster, its identification should be possible using IRAS. Thus we prepare mass-selected $Pt_n$ single-size clusters ($7 \leq n \leq 35$, $n =$ number of Pt atoms in a cluster) on the surface of a well-defined single crystalline oxide support ($Al_2O_3$ or $TiO_2$) and perform IRAS measurements. We also perform the temperature-programmed reaction (TPR) of $CO_2$ to determine how many introduced CO and O molecules are converted to $CO_2$ for various Pt cluster sizes, and thus obtain information about the size and chemical state dependence of the CO oxidation activity.

In this study, we find that the CO oxidation mass activities for clusters with a second Pt layer are higher than those for clusters with a single layer, despite their small number of surface-exposed Pt atoms or large number of embedded atoms. We also show that the highest CO oxidation activity (amount of $CO_2$ produced per deposited $Pt_n$ single clusters. Note that the total number of deposited Pt atoms is the same for each sample; $\leq 0.02$ ML; 1 ML = $1.5 \times 10^{15}$ Pt atoms $cm^{-2}$) is achieved when the ratio of neutral to cationic Pt atoms in the nanocluster is maximized. We detect no correlation between the number of surface-exposed Pt atoms (i.e. the surface area of the Pt cluster) and the amount of $CO_2$ produced. These results are explained by an enhancement of the bond strengths between the Pt atoms located at the edges of the first

cluster layer and the support oxide, which alters the chemical state of these Pt atoms from neutral to cationic, thus weakening the association of O with cationic Pt and resulting in low $CO_2$ production. Furthermore, using the discovered relationship and our modified bond-additivity model (BAM), which uses only the bond energy between a single catalyst atom (in this study, Pt) and the support ($Al_2O_3$ or $TiO_2$) to construct the thermodynamically stable morphologies of the clusters, we show that it is possible to predict the CO oxidation activities of the constructed cluster models.

## Results

**Model structures of Pt clusters on $Al_2O_3$ and $TiO_2$ substrates.** Two well-defined model catalyst systems were prepared by depositing a small amount of mass-selected Pt clusters uniformly on $Al_2O_3$/NiAl(110) and $TiO_2$(110) surfaces without cluster fragmentation and aggregation[20,21]. $Al_2O_3$/NiAl(110) surfaces are used commonly in model catalyst studies because $Al_2O_3$ films grown epitaxially on NiAl(110) exhibit a high degree of crystallinity, low surface roughness, and excellent preparation reproducibility[22,23]. Although $Al_2O_3$ is an insulator, the $Al_2O_3$ film is sufficiently thin to avoid charge accumulation, which enables us to utilize experimental methods involving ions and electrons to characterize the model catalyst systems. The $Al_2O_3$ film contains unsaturated penta-coordinated $Al^{3+}$ cations[22], known as Pt anchor sites on the (100) facets of the $\gamma$-$Al_2O_3$ surface[24,25], which can suppress the thermal diffusion of Pt atoms on the $Al_2O_3$ film. For the $TiO_2$ substrate, the $TiO_2$(110) surface was annealed at ~950 K for 10 min to introduce oxygen vacancies and to ensure electrical conductivity[26].

**CO oxidation activity as a function of Pt cluster size.** Figure 1a shows the procedure employed for studying the CO oxidation activity by the TPR of $CO_2$ formed by the reaction of CO and O (Eq. (1)) on the model structure; first, Pt clusters were saturated by labelled $O_2$ (1000 L of $^{18}O_2$ at 300 K, 1 L = $1 \times 10^{-6}$ Torr·s), followed by saturation adsorption of $^{13}CO$ at 88 K, and finally the TPR measurement was performed[27]. We only detected $CO_2$ with a mass of $m = 47$ regardless of the cluster size (Supplementary Fig. 2), indicating that $CO_2$ is produced from the introduced $^{18}O_2$ and $^{13}CO$ without the involvement of lattice oxygen in $Al_2O_3$, which is consistent with previous reports[13,14].

Figure 1b represents the amount of produced $CO_2$ from sample surface with mass-selected Pt single clusters relative to the number of Pt atoms in the cluster, which was estimated from the area intensity analysis using the TPR spectra shown in Fig. 1c. As clearly seen in Fig. 1b, the amount of produced $CO_2$ peaks at $n = 24$. Figure 1d represents the total amount of unreacted CO estimated from Fig. 1e in the same way. The conversion efficiency from adsorbed CO to $CO_2$ and the amounts of adsorbed CO and O for each cluster size are summarized in Fig. 1f and Supplementary Fig. 3a, b, respectively. Interestingly, $CO_2$ is produced at the same temperature of ~300 K for all cluster sizes (Supplementary Note 1). This result indicates that the same activation barrier exists for each Pt cluster (Supplementary Note 2 and Supplementary Fig. 4). We confirmed that the morphology and size of the Pt clusters were maintained after oxidation at room temperature (Supplementary Fig. 5). Without Pt clusters, no $CO_2$ was produced (Supplementary Fig. 2). The Pt clusters on $Al_2O_3$ were thermally stable at temperatures up to 600 K under ultrahigh vacuum (UHV) conditions (Supplementary Fig. 6), suggesting that the cluster size is maintained in the TPR experiments.

As the number of Pt atoms in the cluster is controlled using a mass filter, the exact number of Pt atoms in each cluster is known. By contrast, the shape of the cluster can only be confirmed by

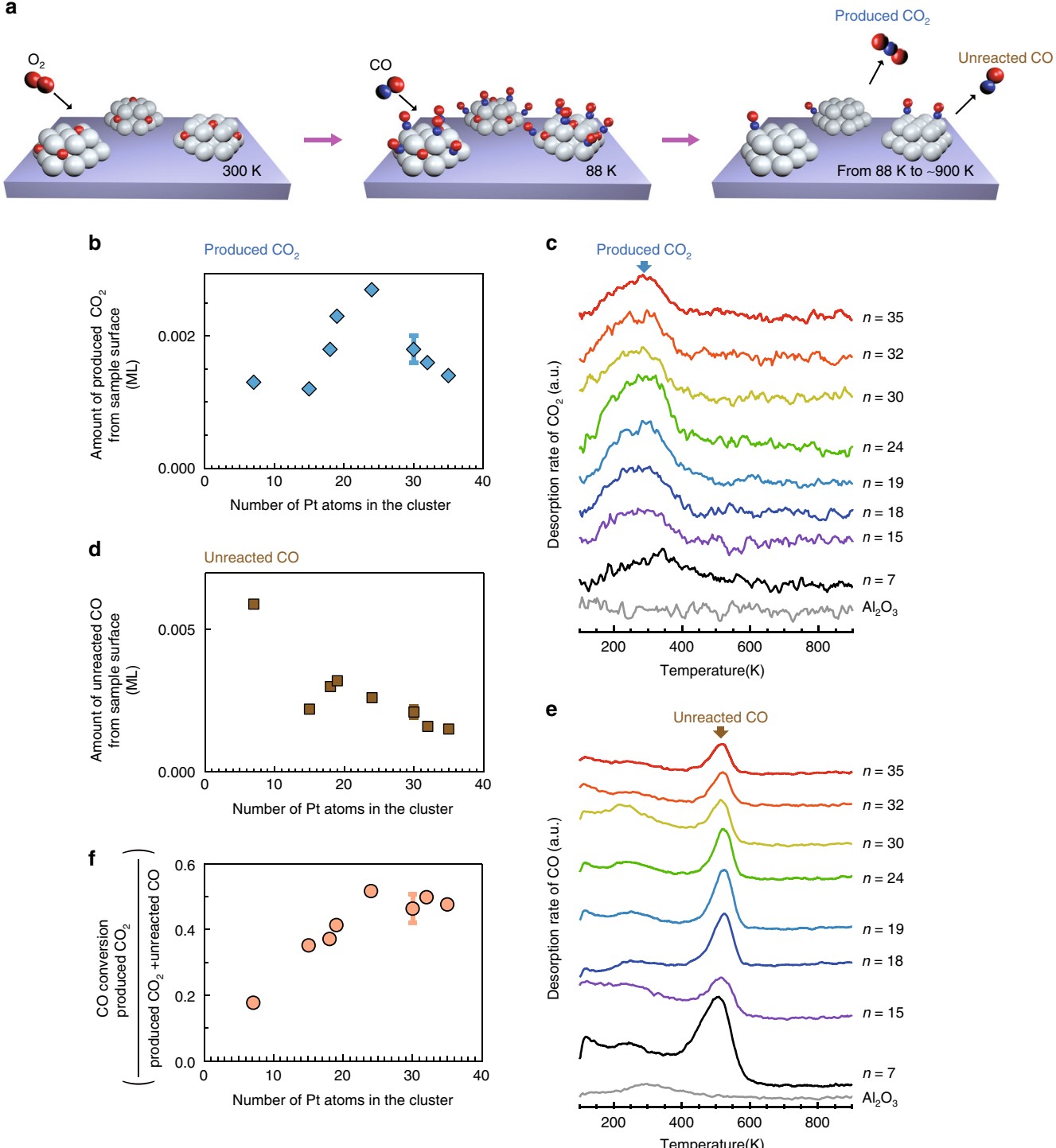

**Fig. 1 CO oxidation activity of mass-selected $Pt_n$ clusters on $Al_2O_3$. a** Overview of the $CO_2$ temperature-programmed reaction (TPR) measurements. The $Pt_n$-deposited surfaces were exposed to 1000 L of $^{18}O_2$ at 300 K to saturate the $Pt_n$ clusters with $^{18}O$ atoms, followed by saturation adsorption of $^{13}CO$ at 88 K, and finally the TPR measurement was performed. The Pt coverage was 0.02 ML. **b** Amount of produced $CO_2$ from sample surface with mass-selected Pt single clusters relative to the number of Pt atoms in the cluster. The error bars were determined from multiple measurements on a single cluster size. **c** $CO_2$ TPR spectra ($m/z = 47$) over $Pt_n/Al_2O_3$. The number of produced $CO_2$ molecules was estimated by integrating the intensity between 100 and 500 K. **d** Amount of unreacted CO molecules relative to the number of Pt atoms in the cluster. **e** CO TPR spectra ($m/z = 29$) over $Pt_n/Al_2O_3$. **f** CO conversion (ratio of produced $CO_2$ molecules to adsorbed CO molecules) as a function of the number of Pt atoms in the cluster. The error bars in **b**, **d**, and **f** are standard deviation. Source data are provided as a Source Data file.

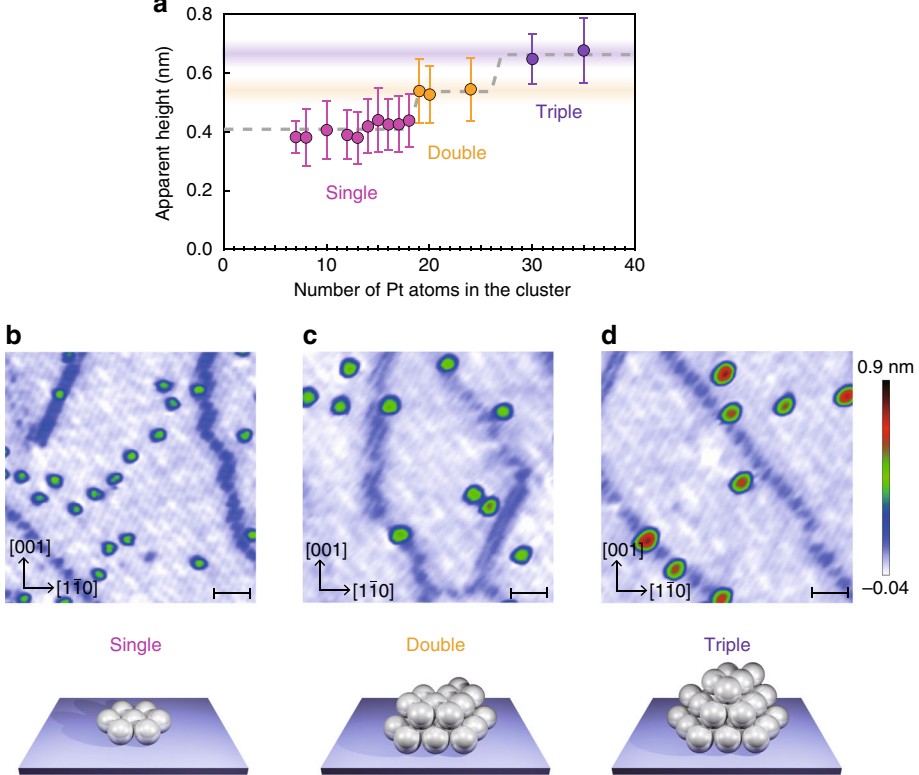

**Fig. 2 Morphologies of mass-selected Pt$_n$ clusters on Al$_2$O$_3$. a** Size dependency of average cluster height on Al$_2$O$_3$. The error bars and dashed lines represent the standard deviations and morphologies simulated using the modified bond-additivity model. Source data are provided as a Source Data file. **b–d** Scanning tunnelling microscopy (STM) images of Pt$_n$/Al$_2$O$_3$ for $n = 7$ (**b**), $n = 24$ (**c**), and $n = 35$ (**d**) (scale bars: 5 nm). The STM measurements were performed at 78 K.

direct imaging[28]. Thus we performed a scanning tunnelling microscopy (STM) analysis of the Pt$_n$ clusters on the Al$_2$O$_3$ support. More than 50 clusters were analysed statistically to estimate the frequency of Pt clusters of different sizes (~2000 clusters in total). The Pt clusters exhibit a single layer up to $n = 18$ and then a double layer is observed up to $n = 30$, at which point a triple layer appears (Fig. 2a; see also Supplementary Fig. 7). Figure 2b–d show representative STM images of Pt$_n$ clusters on the Al$_2$O$_3$ support at $n = 7$, 24, and 35, in which single, double, and triple layers of Pt$_n$ clusters appear. Interestingly, the amount of produced CO$_2$ (Fig. 1b) increases upon changing from single- to double-layer clusters and then decreases when the third layer is added, which indicates an apparent correlation between the CO oxidation activity and the cluster height. However, we observed no correlation between the amount of produced CO$_2$ and the number of surface-exposed Pt atoms in Pt$_n$/Al$_2$O$_3$ (Supplementary Fig. 8).

**Position of cationic Pt atoms within clusters**. Double-layer Pt clusters of ~1 nm in diameter had higher CO oxidation activities than single- and triple-layer clusters on Al$_2$O$_3$. This result is counterintuitive because single-layer clusters have a larger number of surface-exposed Pt atoms than the double- and triple-layer cluster and thus should show higher CO oxidation activities. Therefore, it is likely that some of the Pt atoms in the first layer of the cluster are catalytically less active or have different chemical states.

IRAS analyses have revealed that $^{13}$CO bound to low-coordinated neutral Pt (for example, at the nanocluster edge) exhibits a peak at approximately 2000–2029 cm$^{-1}$ [14,16,29,30], and many studies have confirmed that adsorption on the oxidized species results in a peak at approximately 2032–2081 cm$^{-1}$. Thus

IRAS is suitable for judging the chemical states (we note that an exception has been reported for adsorption on single Pt atoms, resulting in a peak at 2012 cm$^{-1}$, as shown in Supplementary Fig. 9 and Supplementary Table 1).

We collected IRAS spectra for the prepared samples before and after exposure to $^{13}$CO to observe the peak corresponding to the $^{13}$CO stretching mode. This method, known as the CO probe method, is commonly used to investigate the chemical states of Pt[16–19]. It should be noted that at low temperatures (herein, 88 K) CO cannot adsorb on the bare Al$_2$O$_3$ support, only on Pt[31], and that the coordination number and the chemical state of Pt both affect the CO stretching mode.

Figure 3a shows the IRAS spectra for various sizes of Pt clusters on Al$_2$O$_3$. We observed a peak at 2020 cm$^{-1}$ for the Pt$_n$ clusters with $n = 19$–32, indicating the existence of neutral Pt atoms. As the cluster size decreased, a small peak appeared at approximately 2044 cm$^{-1}$ for $n = 19$ and this peak became prominent at $n = 7$. A corresponding decrease in the peak at 2020 cm$^{-1}$ indicated that the ratio of the number of oxidized Pt atoms to the number of neutral Pt atoms increased as the cluster size decreased. The two IRAS peaks observed at 2020 and 2044 cm$^{-1}$ can be attributed to neutral and cationic Pt atoms, respectively, in accordance with the results of previous studies (Supplementary Fig. 9). The IRAS peak for Pt(II) on Al$_2$O$_3$ is known to appear at wavenumbers >2060 cm$^{-1}$ owing to the isotopic shift[17,32]; thus it is reasonable to assume that oxidized Pt bears only a slight positive charge (herein, this Pt is referred to as cationic Pt). To further verify this assignment, we analysed Pt$_7$/Al$_2$O$_3$ using X-ray photoelectron spectroscopy (XPS), which demonstrated that Pt within the Pt$_7$ cluster has an oxidation state between Pt(0) and Pt(II) (Supplementary Fig. 10).

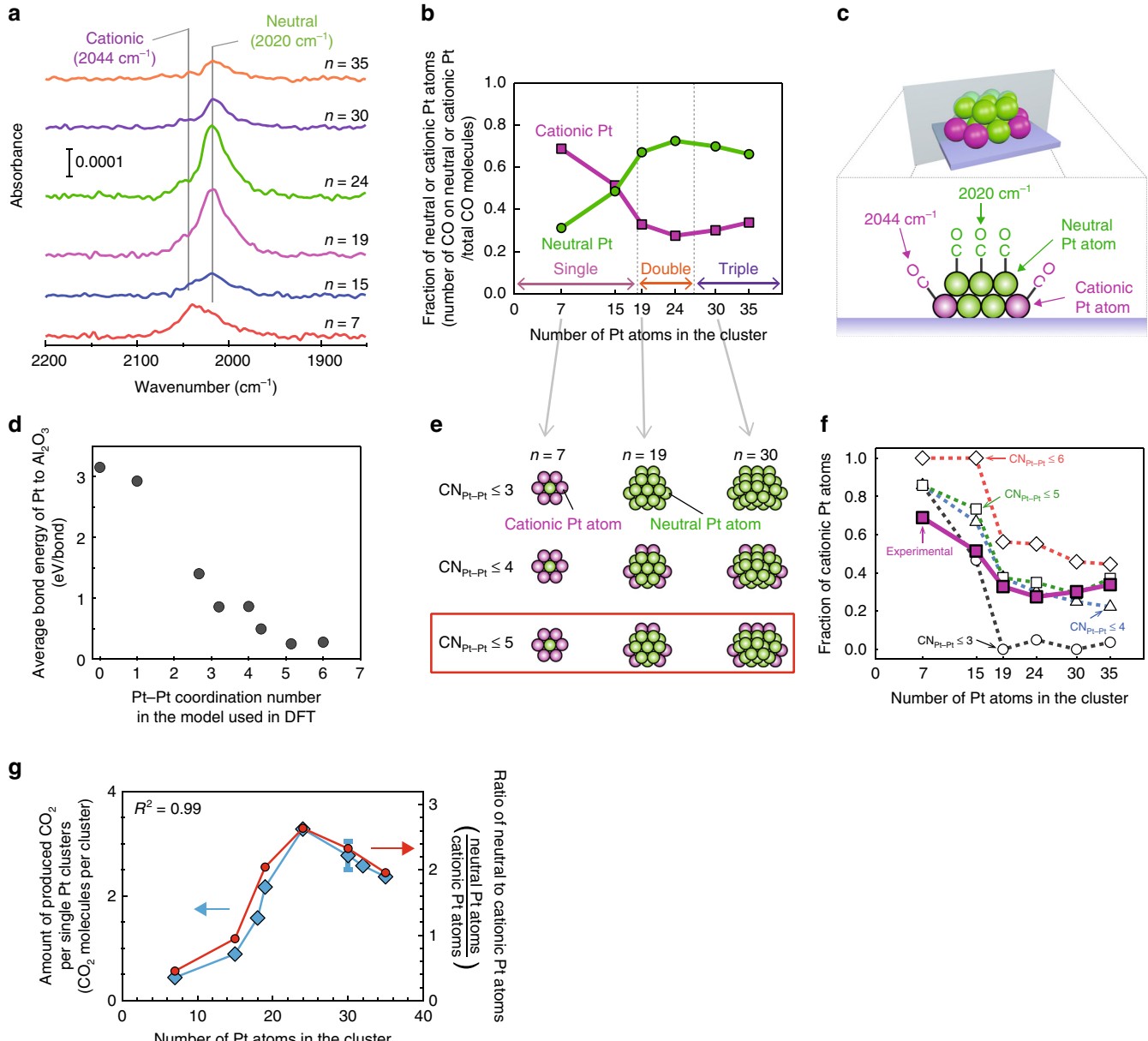

**Fig. 3 Chemical states of Pt and their effect on CO oxidation activity. a** Infrared reflection absorption spectroscopy (IRAS) spectra of $^{13}CO$ adsorbed on $Pt_n/Al_2O_3$. **b** Fraction of neutral and cationic Pt atoms estimated from IRAS spectra. Circles and squares represent fraction of neutral and cationic atoms, respectively. **c** Schematic diagram illustrating the adsorption of CO on a Pt cluster. **d** Average bond energy of $Pt–Al_2O_3$ determined using density functional theory (DFT) calculations. A linear increase in the bond energy between Pt and $Al_2O_3$ is observed as $CN_{Pt–Pt}$ is decreased from 5 to 0. The structural models used for these calculations are shown in Supplementary Fig. 12a. **e** Schematic diagram of clusters showing the location of cationic Pt atoms for different Pt coordination numbers. **f** Fraction of cationic Pt atoms in the cluster. Good agreement with the experimental data (purple squares) is observed for $X = 5$ (open squares). $X = 3$, 4, and 6 are represented as open circles, open triangles, and open diamonds, respectively. **g** Relationship between the amount of produced $CO_2$ per Pt single clusters (circles) and the ratio of neutral to cationic Pt atoms (diamonds). The coefficient of determination ($R^2$) between these parameters was calculated to be 0.99 using the least mean square method. The error bar in **g** is standard deviation. Source data are provided as a Source Data file.

Figure 3b shows the atomic fractions of the two types of Pt atoms as functions of the number of Pt atoms in the cluster (Supplementary Fig. 11 and Supplementary Note 3, which describes the method employed for estimating the atomic fractions). A steep decrease in the atomic fraction of cationic Pt (or increase in the fraction of neutral Pt) is observed in the single-layer cluster region, and this decrease levels off in the double- and triple-layer cluster regions. This result suggests that cationic Pt atoms exist predominantly in the first layer of Pt clusters, presumably at the perimeter of the cluster (Fig. 3c). To elucidate

the exact position of cationic Pt atoms within the clusters, we hypothesized that under-coordinated Pt atoms bind to the support surface more strongly and consequently become cationic. This suggestion is reasonable because the coordination number of adsorbates has been shown to affect the adsorption energy[33] and because charge transfer from Pt to $Al_2O_3$ has been demonstrated[34]. To verify this hypothesis, we theoretically calculated the total binding energy of each Pt cluster on $Al_2O_3$ (Supplementary Fig. 12 and Supplementary Note 4). As hypothesized, these calculations revealed that a lower coordination number for the Pt

atom results in a stronger bond with $Al_2O_3$ (Fig. 3d). This tendency can be understood intuitively from the principle of bond order conservation[35] and can also be explained from the results reported for Ag on MgO(100), where the adhesion energy (or bond energy) increased with decreasing Ag particle size[36].

The atomic fraction of cationic Pt atoms with respect to the number of Pt atoms in the cluster (Fig. 3b) should match well with that obtained from the model in which the neutral and cationic Pt atoms were correctly assigned by assuming that the intensities of the IRAS peaks at 2020 and 2044 $cm^{-1}$ are proportional to the number of neutral and cationic Pt atoms, respectively. In other words, if we can develop a model that matches well with the IRAS results, we can deduce the coordination number that results in the atoms becoming cationic. To clarify this point, we constructed several structural models by manually assigning which Pt atoms were cationic. We assumed that all Pt atoms in the cluster were cationic if the coordination number of Pt to neighbouring Pt atoms ($CN_{Pt-Pt}$) was below a certain number, $X$ (Fig. 3e), and searched for the $X$ that provided a match to the experimental IRAS data. At $X = 5$, the curves matched the IRAS data reasonably well (Fig. 3f), which indicated that Pt atoms with $CN_{Pt-Pt}$ values < 5, located at the perimeter of the cluster, behaved as cationic atoms.

**Effect of Pt chemical states on CO oxidation activity**. The analysis performed thus far revealed two important findings. First, there is no correlation between the amount of produced $CO_2$ and the number of surface-exposed Pt atoms in $Pt_n/Al_2O_3$. Second, cationic Pt appears at the edges of the first layer of the Pt clusters and the number of cationic Pt atoms increases with a decrease in the Pt cluster size.

Given that CO molecules adsorb on both neutral and cationic Pt atoms, the total amount of adsorbed CO should be proportional to the total number of surface-exposed neutral and cationic Pt atoms ($\theta_{CO} \propto N_n + N_c$). As O has a high electronegativity, it is reasonable to assume that O prefers to adsorb on neutral Pt atoms ($\theta_O \propto N_n$). Thus the ratio of the amount of adsorbed CO to O, that is, $\theta_{CO}/\theta_O$, can be expressed by Eq. (2):

$$\theta_{CO}/\theta_O \propto (N_n + N_c)/N_n = 1 + N_c/N_n \qquad (2)$$

In fact, $\theta_{CO}/\theta_O$ was matched well with $(N_n + N_c)/N_n$ as shown in Supplementary Fig. 3c. For CO oxidation, equivalent amounts of CO and O would result in the CO oxidation activity being maximized when $N_c/N_n$ is zero. In other words, the Pt cluster with the maximum $N_n/N_c$ value will show the highest CO oxidation activity, and thus it will produce the largest amount of $CO_2$. Figure 3g shows the ratio of neutral to cationic Pt atoms ($N_n/N_c$) estimated by IRAS and the amount of produced $CO_2$ measured using TPR as functions of the number of Pt atoms in the cluster. As expected, we found a strong correlation between the $N_n/N_c$ and the amount of produced $CO_2$ (the $R^2$ value estimated by the least mean square method was 0.99). This observation suggests that cationic Pt has a low oxygen affinity, which explains why the smallest cluster does not show the highest CO oxidation activity in this study.

This strong correlation would be practically useful for estimating the CO oxidation activity for different cluster sizes if the thermodynamically most stable Pt atomic arrangement could be determined for each cluster size and enabled to calculate $N_n$ and $N_c$.

**Thermodynamically stable Pt cluster morphologies**. The most stable arrangement can be determined by searching for the cluster model that gives the highest binding energy ($E_{bd}$), as calculated by summing the bond energies of each Pt–Pt and Pt–support bond (Eq. (3))[36–38].

$$E_{bd} = \sum_{i=1}^{N_{Pt-Pt}} E_{Pt-Pt,i} + \sum_{j=1}^{N_{Pt-S}} E_{Pt-support,j}, \qquad (3)$$

where $E_{Pt-Pt,i}$ and $E_{Pt-support,j}$ are the bond energies of $i$th Pt–Pt bond and the $j$th Pt–support bond, respectively. $N_{Pt-Pt}$ and $N_{Pt-support}$ represent the numbers of Pt–Pt bonds and Pt–support bonds, respectively (Fig. 4a). This method is known as BAM. Although the conventional BAM assumes constant bond energies for both Pt–Pt and Pt–support bonds, an enhancement in bond strength from the bulk to under-coordinated Pt atoms ($CN_{Pt-Pt} \leq 5$) should be taken into account. Thus we implemented this enhancement into the BAM as follows. First, all the possible cluster models that exhibited individual single, double, triple, and quadruple layers for different cluster sizes were constructed (Fig. 4b). Second, the binding energies for all the models were calculated by summing the Pt–support bond energies for individual Pt atoms that have different coordination numbers in the cluster (Fig. 4c). Finally, the binding energy of each model was plotted as a function of the number of Pt atoms in the cluster (Fig. 4d), and the transitions from single to double and from double to triple layers were determined.

To estimate the binding energy, it is necessary to calculate all the individual bond energies in the cluster; for this purpose, the linear function shown in Supplementary Eq. (6), deduced by density functional theory (DFT) (Fig. 3d), was used. We assumed that the bond energy changed linearly as a function of the coordination number of the Pt atoms in the cluster. Consequently, two appropriate bond energies for $CN_{Pt-Pt} = 0$ and $CN_{Pt-Pt} = 5$ were assigned (Fig. 4c), and the slope and the intercept of Supplementary Eq. (6) were calculated. Subsequently, the coordination number of the Pt atom of interest in the cluster was incorporated into the equation, allowing determination of its bond energy. We repeated this process for all remaining Pt atoms in the cluster by assigning different bond energies at $CN_{Pt-Pt} = 0$ and 5, until the transitions from single to double and from double to triple layers appeared at $n = 18$–19 and at $n = 24$–30 (Fig. 4d), respectively, as observed by STM (Fig. 2a).

Consequently, we determined the binding energy of the Pt monomer, corresponding to $CN_{Pt-Pt} = 0$, to be 3.9 eV, whereas that for $CN_{Pt-Pt} = 5$ was determined to be 1.0 eV (~30% of the bond energy at $CN_{Pt-Pt} = 0$). For the Pt–Pt bond, the bulk Pt–Pt value of 0.98 eV was used (see Supplementary Notes 5 and 6 and Supplementary Figs. 13–15). The reported binding energy of the Pt monomer calculated by DFT is 3.5 eV[25], which is in good agreement with our value (3.9 eV).

Accordingly, we calculated the bond energy of each Pt–Pt and Pt–support bond with a different coordination number and constructed the most thermodynamically stable morphology for each cluster size.

**Prediction of the CO oxidation activity of Pt clusters**. Using the discovered relationship and our modified BAM method (Fig. 5a), it should be possible to predict the CO oxidation activities of different cluster sizes. Thus we first built the cluster models using the modified BAM. Two bond energies were used to consider the bond enhancement effect: 3.50 eV for $CN_{Pt-Pt} = 0$, as previously calculated using DFT[25], and 1.05 eV (30% of 3.50 eV) for $CN_{Pt-Pt} = 5$, as calculated using Supplementary Eq. (6) for $Pt_n/Al_2O_3$. Next, we calculated the ratio of neutral to cationic Pt atoms using these cluster models (Fig. 5b). The results were in excellent agreement with the amounts of $CO_2$ produced for $Pt_n/Al_2O_3$, which displayed a peak at a cluster size of $n = 24$, confirming that it is possible to predict the optimum Pt cluster size for maximizing the CO oxidation activity.

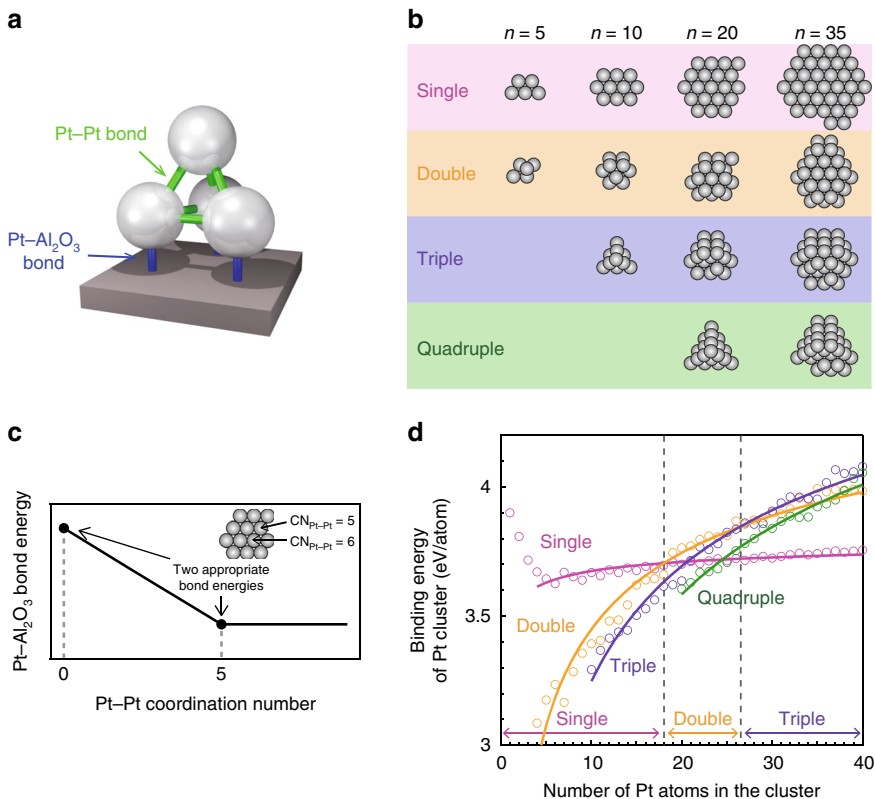

**Fig. 4 Simulation of cluster morphologies using the bond-additivity model. a** Schematic illustration of a double-layer $Pt_4$ cluster. The Pt–Pt and Pt–$Al_2O_3$ bonds are represented using green and blue lines, respectively. **b** Top view of the models for $Pt_n$ structures for which binding energies were calculated using the bond-additivity model (BAM). The numbers of Pt–Pt and Pt–$Al_2O_3$ bonds were counted based on these model structures. **c** Schematic illustration of the Pt–$Al_2O_3$ bond energy as a function of the Pt–Pt coordination number, which was used to calculate the binding energies of the $Pt_n$ clusters using the BAM. The arrows indicate the bond energies at $CN_{Pt–Pt} = 0$ and 5, which were used to calculate the slope and intercept of the linear function. **d** Binding energies of $Pt_n$ clusters with single, double, triple, and quadruple layer (represented by pink, orange, blue, and green circles, respectively) calculated using the BAM as a function of cluster size. The two vertical dashed lines show the simulated sizes corresponding to morphological transitions from single to double layers ($n = 18$-19) and from double to triple layers ($n = 26$-27). Source data are provided as a Source Data file.

To further verify the feasibility of this method, which only uses the appropriate bond energy between the Pt monomer and the support, we estimated the CO oxidation activities for the Pt nanoclusters on a $TiO_2$ surface. To construct the thermodynamically stable structures with our modified BAM, we used 2.4 eV, the reported bond energy between Pt monomers and $TiO_2$[39], as the bond energy for $CN_{Pt–Pt} = 0$ and 0.72 eV for $CN_{Pt–Pt} = 5$, which is 30% of the bond energy at $CN_{Pt–Pt} = 0$. The bond energy for $CN_{Pt–Pt}$ values between 2 and 4 were linearly interpolated. Similar to the $Pt_n/Al_2O_3$ case, the calculated results showed a very good correlation with the experimentally obtained amounts of produced $CO_2$ for $Pt_n/TiO_2$ (Fig. 5c and Supplementary Fig. 16). The clusters with 15–30 atoms, corresponding to diameters of ~0.8–1.4 nm, showed high CO oxidation activities. Such good agreement was not achieved without assuming the existence of cationic Pt, indicating that cationic Pt atoms also exist in the clusters on $TiO_2$. This finding is in line with the previous observation that the Pt 4f XPS peak is positioned between the peaks corresponding to oxidized Pt(II) and neutral Pt(0)[11].

## Discussion

In summary, the CO oxidation activities of mass-selected Pt nanocluster catalysts on two typical metal oxide supports were investigated. It was shown that Pt atoms become cationic when their coordination number to neighbouring Pt atoms is <5. Our IRAS analyses identified two different peaks corresponding to neutral and cationic Pt, the latter of which bears a slightly positive charge. These cationic Pt atoms appear only in the first layer of the cluster and benefit from an enhancement in binding to the support surface. Although no correlation was observed between the CO oxidation activity and the total number of surface-exposed Pt atoms, a strong correlation was found between the atomic ratio of the neutral to cationic Pt atoms in the cluster ($N_n/N_c$) and the amount of produced $CO_2$ for $Pt_n/Al_2O_3$. We exploited this strong correlation to predict the cluster size that produces the largest amount of $CO_2$ for $Pt_n/TiO_2$ by simulating the most stable morphologies for clusters with different numbers of atoms using our modified BAM and calculating the number of neutral and cationic Pt atoms manually. The simulated atomic ratio of the neutral to cationic Pt atoms was in good agreement with the amounts of produced $CO_2$ for $Pt_n/TiO_2$ estimated experimentally from TPR measurements. From the point of view of utilization, smaller catalysts would be expected to perform better than larger ones; however, our findings revealed that this expectation is not always true because the substrate material affects catalysts with diameters in the sub-nanometre to few atoms range owing to the existence of under-coordinated atoms at the perimeter of the cluster. We note that differences exist between the catalysis conditions in practical and model catalyst systems; for example, the oxide support is usually large and non-conducting in real catalyst systems. Furthermore, the cluster morphology might change dynamically during the reaction, as observed in several recent studies[40,41]. Nonetheless, we believe

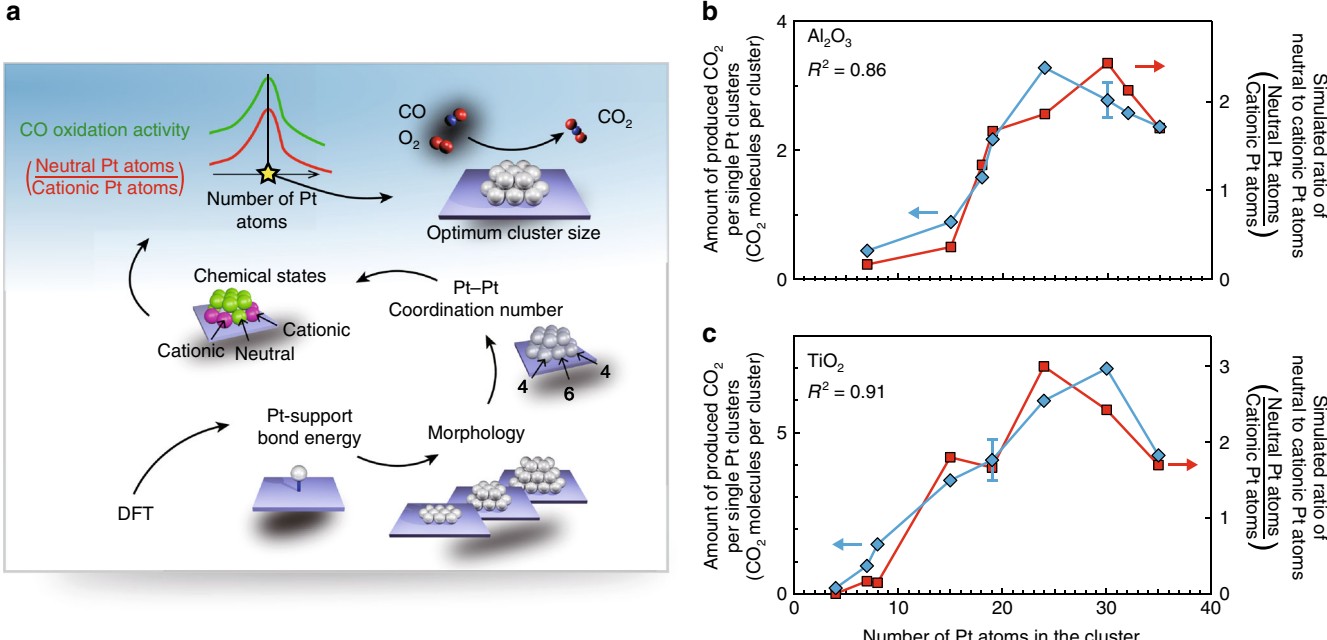

**Fig. 5 Prediction of the optimum Pt cluster size for maximum CO oxidation activity. a** Schematic diagram illustrating the protocol used for optimum Pt cluster size prediction. The process starts with a theoretical calculation of the binding energy for a Pt monomer on the support material. **b, c** Comparison of the atomic ratios of neutral to cationic Pt atoms (red squares) with the amounts of $CO_2$ produced for $Pt_n$/$Al_2O_3$ (**b**) and $Pt_n$/$TiO_2$ (**c**) (blue diamonds). The error bars are standard deviation. Source data are provided as a Source Data file.

that the results presented in this study will be helpful for designing atomically dispersed nanostructured catalysts including single-atom and few-atom clusters or single-cluster catalysts, which have recently received considerable interest[5,42–45].

## Methods

**Sample preparation.** All experiments were performed in a UHV chamber (<1 × $10^{-7}$ Pa)[20]. NiAl(110) substrates were oriented to within 0.1° and 10 mm in diameter for the STM experiments and were 6 mm × 10 mm in size for the TPR and IRAS experiments (Surface Preparation Laboratory). For the IRAS and TPR experiments, the temperature was monitored by a chromel–alumel ($K$-type) thermocouple that was spot-welded to the side of the substrate. The clean NiAl(110) surface was prepared by several cycles of $Ar^+$ sputtering (300 K, 20 min) and annealing at 1300 K for 5 min. The thin $Al_2O_3$ film was prepared by dosing 1800 L (Langmuir: 1 × $10^{-6}$ Torr·s) of $O_2$ at 600 K, followed by annealing at 1100 K for 5 min[23]. The oxidation process was repeated more than three times to close open NiAl(110) patches in the film. We confirmed that the $Al_2O_3$ film covered the NiAl(110) surface fully by STM measurements. In addition, no IRAS peak characteristic of CO adsorbed on NiAl(110) was observed after exposure of the $Al_2O_3$/NiAl(110) surface to CO at 88 K, clearly showing the absence of bare NiAl(110) regions. $Al_2O_3$ film showed a characteristic IRAS peak corresponding to a crystalline $Al_2O_3$ film without amorphous components[46], which is consistent with the observation of the low energy electron diffraction pattern of the crystalline $Al_2O_3$ surface[23] with sharp spots and a low background intensity. The above data clearly show that our films are crystalline and fully cover the NiAl(110) surface.

The $TiO_2$(110) substrates, 10 mm × 10 mm in size, were oriented to within 0.5° for the STM experiments; for the TPR experiments, substrates of 6 mm × 10 mm in size were used (Sinkosha Co., Ltd). For the TPR experiments, the temperature was monitored by a $K$-type thermocouple that was inserted into a slot at the side of the substrate with a Ta foil to ensure good thermal contact. The clean $TiO_2$(110) surface was prepared by several cycles of $Ar^+$ sputtering (300 K, 10 min) and annealing at 900–980 K for 10 min. After the annealing process, the temperature was decreased to 300 K at a cooling rate of 1 K $s^{-1}$. Based on the magnitude of the OH recombination peak in the temperature-programmed desorption (TPD) spectrum of $H_2O$, the coverage of surface oxygen vacancies was estimated to be 8% of the number of surface $Ti^{4+}$ atoms (5.2 × $10^{14}$ atoms $cm^{-2}$), which is a typical value for standard surface preparation under UHV[47].

$Pt_n$ cluster ions were produced by a DC magnetron sputtering source (Angstrom Sciences, ONYX-1) and mass selected using a quadrupole mass filter (Extrel, MAX-16000) by sputtering a Pt target (Tanaka Kikinzoku Kogyo K.K., purity > 99.99%)[20]. Mass-selected $Pt_n$ cluster ions were uniformly deposited on the surfaces at 300 K using the Lissajous scanning method[48]. The impact energy for

cluster deposition was tuned to <2 eV $atom^{-1}$ (soft-landing condition) by adjusting the bias voltage applied to the surface, where the impact energy was estimated using the retarding voltage method. The total amount of Pt deposited was determined from the integrated $Pt_n^+$ neutralization current on the sample and was confirmed using STM and XPS. Small amounts of $Pt_n$ clusters were deposited to avoid cluster aggregation in the deposition process. The Pt coverage was 0.02 ML (1 ML = 1.5 × $10^{15}$ atoms $cm^{-2}$) for the TPR and IRAS experiments, and <0.02 ML for the STM experiments, where >90% of the deposited clusters were adsorbed without aggregation[21].

**TPR experiments.** To investigate the catalytic reaction of adsorbed species on the $Pt_n$ clusters, we used isotopically labelled $^{18}O_2$ (Sigma-Aldrich Co., isotopic purity = 97%) and $^{13}CO$ (Cambridge Isotope Laboratories, Inc., isotopic purity = 99%). These gases were introduced through a pulse gas dosing system onto the sample surface, and the purity of each gas was checked using mass spectrometry. For the TPR experiments, the model catalyst samples were first exposed to 1000 L of $^{18}O_2$ at 300 K to saturate the $Pt_n$ cluster surface by atomic oxygen, followed by saturation adsorption of $^{13}CO$ at 88 K. Since the temperature in a catalytic converter in diesel combustion engines is typically above ~550 K, and the concentration of $O_2$ is much higher than that of CO molecules, which desorb from the Pt catalyst surface at ~500 K, the surface of the Pt catalyst would be covered by $O_2$ during CO oxidation. To mimic this situation, we pre-adsorbed $O_2$ before adsorbing CO, and we investigated the catalyst activity by TPR measurements. During the TPR measurements, the sample temperature was ramped at a rate of 3.5 K $s^{-1}$ and desorbed molecules were detected using a quadrupole mass spectrometer (QMS, Pfeiffer, PrismaPlus QMG220M1). The ionization region was enclosed in a small glass envelope (Feulner cup) with a small opening (3 mm in diameter), and the crystal surface was placed in front of this opening at a distance of 1 mm. The mass spectrometer sensitivity was calibrated by measuring the TPD spectrum of (2 × 2) O/Pt(111) for every experiment. The area intensities for CO and $CO_2$ coverages were analysed as follows: we collected ≥10 data points measured from ≤100 K and ≥500 K to determine the background level (Supplementary Fig. 17 and Supplementary Note 7). The CO and $CO_2$ coverages were estimated by comparing the area intensities of the TPD and TPR spectra with those of c(4 × 2)CO/Pt(111) (0.50 ML of CO) and CO+O/Pt(111) at the saturation coverage (0.25 ML of $CO_2$). Background $^{12}CO$ could be adsorbed on the Pt clusters if the prepared sample is not swiftly investigated and if the surface is exposed for a long time in the UHV chamber. It should be possible to estimate the amount of $^{12}CO$ molecules to which the sample was exposed based on the time of exposure to background $^{12}CO$ (or the time required to prepare the samples and record the IRAS and TPD measurements, which is ~77 min) and its partial pressure (~1.7 × $10^{-9}$ Pa). In our experiment, the $^{12}CO$ exposure amount was estimated to be <0.1 L (1 L = 1.33 × $10^{-4}$ Pa·s), which is quite low.

To quantify the amount of background $^{12}CO$ adsorbed on the Pt clusters, we exposed the Pt-cluster-deposited $Al_2O_3$ to $^{13}CO$ and performed TPD and IRAS measurements for both $^{12}CO$ and $^{13}CO$ (Supplementary Fig. 18). Adsorbed $^{12}CO$ and $^{13}CO$ were observed to desorb at ~500 K, and by integrating the intensity of their peaks, we confirmed that the fraction of $^{13}CO$ was >0.9 for all the prepared Pt cluster sizes, as shown in Supplementary Fig. 18b. Thus we conclude that the effect of adsorbed background $^{12}CO$ is minimal, which is also supported by the IRAS spectra, in which the stretching-vibrational modes of $^{12}CO$ at 2090 and 2066 cm$^{-1}$ were not observed, as indicated in grey in Supplementary Fig. 18c.

We detected no $O_2$ desorption during the TPR measurements, as in a previous report[49]. Thus, based on the previous report, the O coverage was calculated by assuming that all the adsorbed O was consumed by CO during the reaction (Supplementary Fig. 3)[49].

Desorption rates of CO and $CO_2$ from the sample exposed to $^{13}CO$ and $^{18}O_2$ were estimated by measuring the corresponding ion currents using QMS, whereby the sample surface was heated at a constant rate (3.5 K s$^{-1}$) and ion current is continuously monitored.

**Scanning tunnelling microscopy.** STM measurements were performed at 78 K using a low-temperature scanning tunnelling microscope (Omicron GmbH) with a Nanonis controller (SPECS Zurich GmbH) and electrochemically etched tungsten tips. The STM images of $Pt_n/Al_2O_3$ were taken at a positive sample bias ($V_s$) of 3.5 V and a tunnelling current ($I_t$) of 0.1 nA and those of $Pt_n/TiO_2$ were taken at $V_s$ of 1–3 V and $I_t$ of 0.05–0.1 nA. To estimate the height of the sample from the STM measurements, we calibrated the height with respect to a single atomic step on NiAl under the same scanning conditions (see also Supplementary Fig. 7b and Supplementary Note 8).

**Infrared reflection absorption spectroscopy.** IRAS measurements were performed using a Fourier-transform infrared spectrometer (Bruker IFS66v/S) with a mercury–cadmium–telluride detector. The incident beam was passed through a KRS-5 polarizer to remove the unwanted s-polarized component. All spectra were collected at a 4 cm$^{-1}$ resolution over 200 scans. IRAS spectra were recorded at a sample temperature of 88 K.

**X-ray photoelectron spectroscopy.** XPS measurements were performed at 300 K using Mg Kα (1253.6 eV) radiation, along with a hemispherical energy analyser (Omicron EA125HR) and a seven-channel detector. XPS spectra were collected at an electron take-off angle of 40° with a pass energy of 20 eV.

**First-principles calculations.** First-principles calculations for $Pt_n/Al_2O_3$ were performed using the projector-augmented wave method based on DFT implemented in the Vienna ab initio simulation package[50]. The generalized gradient approximation of the Perdew–Burke–Ernzerhof (PBE) functional was applied to the exchange–correlation energy[51]. The cut-off energy for the plane waves was set to 500 eV. The integral over the Brillouin zone was approximated by summation on a 3 × 3 × 1 Monkhorst–Pack k-point mesh, and Gaussian smearing with a width of 0.2 eV was applied. Optimization of the crystal structures was performed until the atomic forces were <0.01 eV Å$^{-1}$. The γ-$Al_2O_3$(001) surface was modelled as a slab of a p(1 × 1) cell. The crystal structure reported by Digne et al. was used for the bulk of γ-$Al_2O_3$[52]. The vacuum layer between slabs was set to 15 Å (excluding the Pt cluster).

First-principles calculations on freestanding Pt particles were carried out using linear combinations of pseudo-atomic orbitals with the double zeta plus polarization quality as a basis set and norm-conserving pseudopotentials as effective core potentials[53]. The computational parameters were the same as those used in our previous study[53]. Each NP was located in vacuum, and its structure was optimized to achieve maximum atomic forces of <0.05 eV Å$^{-1}$. The exchange–correlation interactions among electrons were described using the revised PBE functional proposed by Hammer et al.[54].

## Data availability

The source data underlying Figs. 1b–f; 2a; 3a, b, d, f, g; 4d; and 5b, c and Supplementary Figs. 2; 3a–c; 4a–h; 5e–h; 6g–i; 9b; 10; 11a–c; 12b, c; 14a–d; 15a, c, d; and 16a, e are provided as a Source Data file. Extra data are available from the corresponding author upon reasonable request.

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

## Acknowledgements

The authors thank Dr N. Isomura (Toyota Central R&D Labs, Inc.) for STM measurements.

## Author contributions

A.B. conceived and designed the experiments and performed the TPR, IRAS, STM, and XPS measurements. H.H. and Y.W. designed the apparatus. N.O. performed the DFT calculations, and N.O and R.J discussed the results. S.H. conceived the prediction method for the CO oxidation activity of nanoclusters. A.B. and S.H. prepared the manuscript. All authors discussed the results and commented on the manuscript.

## Competing interests

The authors declare no competing interests.
