## [Peer Review File · Nature Communications]

Reviewers' comments:

Reviewer #1 (Remarks to the Author):

The authors performed a combined experimental and computational study on CO oxidation reaction on a series of Pt_n/Al₂O₃ systems (and partially extended to Pt_n/TiO₂). They find that the undercoordinated Pt atoms located at the very edges of the first cluster layer are more positively charged (aka cationic) due to direct contact with the Al₂O₃ support. The overall CO oxidation activity is found to correlate with $\frac{N_{\text{neutral}}}{N_{\text{cation}}/N_{\text{total}}}$, “the ratio of neutral to cationic Pt atoms normalised by the total number of Pt atoms”. Using a bond-additivity model (BAM) that based on computationally or empirically estimated bond energies between a single catalyst atom and the support, they constructed the thermodynamically stable morphologies of the clusters, and predicted the CO oxidation activities of such clusters.

Overall, I think the experimental results of the CO oxidation activity on a series of size-selected Pt_n cluster on Al₂O₃ and TiO₂ surfaces provide interesting data for constructing a meaningful model in predicting the catalytic properties. I am not an expert on the experimental aspects, thus my comments are mainly on the computational results. Many of the findings in this work are consistent with the general understanding of catalysts with nano-particles, thus are not surprising or completely new. For example, the findings that the interfacial atoms are more ionic and the oxidation state of Pt increases with the particle size decreases are such cases. However, they provide clear evidence that the CO oxidation activity is correlated with the ratio $\frac{N_{\text{neutral}}}{N_{\text{cation}}/N_{\text{total}}}$, which is interesting and provides insight for understanding the role of various metal sites with different oxidation states or charge states. I therefore think the manuscript might become publishable for Nature Communications after taking into account of the following comments and suggestions:

- 1) When Pt_n clusters are supported on oxides, the difference of the chemical potentials decides how much amount of charges are transferred, as have been discussed in the literature (e.g. J. Am. Chem. Soc., 2017, 139, 6190). They need to clarify this point because when the metal cluster and the support have similar chemical potentials, there is no cationic atoms, even at the interfacial region.
- 2) The undercoordinated, interfacial Pt atoms are more positively charged if the charge transfer flows from metal to support. Since the authors performed DFT calculations, I am surprised that they did not provide the Bader charges for these atoms at

different layers of the cluster. The charges should be included to provide more direct data on the “chemical state” change of the Pt atom.

3) Conceptually, valence state, oxidation state, and atomic charge are quite different concepts. The authors should avoid mixing them together. The high oxidation state metal (e.g. Pt(II) or Pt⁺²) often carries more positive charge than a low oxidation state metal (e.g. Pt(0)). But one cannot confuse oxidation state with charge state. In

p. 7, “Pt within the Pt₇ cluster has an oxidation state between Pt⁰ and Pt²⁺”. Here Pt²⁺ is the charge state, while Pt(II) or Pt+2 is the oxidation state.

4) The key finding of CO oxidation activity to correlate with $\frac{N_{\text{neutral}}}{N_{\text{cation}}/N_{\text{total}}}$ needs some further explanation of its physical meaning. In fact, $\frac{N_{\text{neutral}}}{N_{\text{cation}}/N_{\text{total}}} = \frac{N_{\text{neutral}}}{(N_{\text{cation}}/N_{\text{total}})} = (N_{\text{total}}/N_{\text{cation}})/(N_{\text{cation}}/N_{\text{total}}) = 1/N_{\text{cation}} + 1/N_{\text{total}}$; Because the N_{total} is usually far larger than N_{cation} , one can imagine that this simply says that the CO oxidation activity is inversely proportional to the N_{cation} . On the other hand, when making this correlation of CO oxidation activity correlating with $\frac{N_{\text{neutral}}}{N_{\text{cation}}/N_{\text{total}}}$, have they excluded those atoms that are inaccessible (e.g. not on the surface) for CO oxidation?

5) The manuscript correlates the cationic state with the coordination number. In fact, both the coordination number and the bond distances matter for the bond energy, as discussed in Nature Chem. 2015, 7, 403. The authors missed this important work.

6) One should realize that the metal clusters might be dynamically changed in structure when CO and O₂ are adsorbed and react (e.g. Nature Commun., 2015, 6, 6511). The intrinsic structure of Pt_n/support may not be the same as it started before the reaction. This should be pointed out in the manuscript to readers.

7) Well-defined clusters embedded or supported on oxides surface provide opportunity for precision control of catalytic reactions. This kind of “single-cluster catalysts” are widely studied lately and should be discussed in the introduction or as perspective.

8) There are some typos, for instance, p. 9 “high electron negativity” should be “high electronegativity”; Pt-support interaction energy should be called “binding energy” not “adsorption energy”. The latter is reserved for molecules (e.g. CO, O₂, CO₂) that adsorb and desorb on the Pt_n/support surface.

Reviewer #2 (Remarks to the Author):

This is a high-quality manuscript on the CO Oxidation reaction, where the authors try to provide a structure-activity relationship. Such knowledge is important and may be useful for purposeful catalyst design. Nevertheless, there are several issues requiring clarification.

- i) A volcano-type dependence of catalyst activity on the number of Pt atoms in different clusters was established (Fig. 1b, Fig.3 g,h and Fig. 5 b,c). It is, however, not correct to relate the activity to the total number of Pt sites for the catalysts with 3-D structured Pt species. In addition, it is not clear why the ratio of neutral to cationic Pt sites should be related to the total number of Pt atoms (See the right Y axis in Fig.3 h and Fig. 5 b,c).
- ii) To support their conclusions, the authors should report surface coverage by adsorbed carbon monoxide and oxygen species before starting the reaction. Can the authors exclude the fact that the ratio of adsorbed carbon monoxide to oxygen species depend on the size of Pt clusters? What is about the kind of adsorbed oxygen species? Does oxygen exist in molecular or atomic forms or their mixtures?
- iii) The authors reported desorption profiles of CO in Fig. 1d. What is about oxygen?
- iv) A weak point of the experimental part of this manuscript is that the authors do not provide the rate of CO oxidation. Can the authors calculate such values from their data?
- v) Will the authors observe the same activity-size dependence under steady-state conditions?

Responses to the comments from reviewers

CO oxidation activity of non-reducible oxide-supported mass-selected few-atom Pt nanoclusters

Atsushi Beniya^{1,*}, Shougo Higashi^{1,*}, Nobuko Ohba¹, Ryosuke Jinnouchi¹, Hirohito Hirata² & Yoshihide Watanabe¹

¹Toyota Central R&D Labs, Inc., 41-1 Yokomichi, Nagakute, Aichi 480-1192, Japan

²Toyota Motor Corporation, 1200 Mishuku, Susono, Shizuoka 410-1193, Japan

We thank the reviewers for their comments and insightful remarks. In this point-by-point response letter, we respond to these comments based on the experimental and calculated data in manuscript, including new information that has been added in response to the reviewers. All the points made here are reflected in the revised manuscript.

The reviewers' comments are shown in ***italic bold*** text. In our responses, the text shown in red indicates added or revised sentences. Significant changes have been made in the manuscript text, including updated figure numbering; thus, in this letter, we refer to figures using the numbering in the revised manuscript. The references cited in this letter are listed at the end of this document.

Reviewer #1 (Remarks to the Author):

The authors performed a combined experimental and computational study on CO oxidation reaction on a series of Pt_n/Al₂O₃ systems (and partially extended to Pt_n/TiO₂). They find that the undercoordinated Pt atoms located at the very edges of the first cluster layer are more positively charged (aka cationic) due to direct contact with the Al₂O₃ support. The overall CO oxidation activity is found to correlate with Nneutral/Ncation/Ntotal, "the ratio of neutral to cationic Pt atoms

normalised by the total number of Pt atoms". Using a bond additivity model (BAM) that based on computationally or empirically estimated bond energies between a single catalyst atom and the support, they constructed the thermodynamically stable morphologies of the clusters, and predicted the CO oxidation activities of such clusters. Overall, I think the experimental results of the CO oxidation activity on a series of size selected Pt_n cluster on Al₂O₃ and TiO₂ surfaces provide interesting data for constructing a meaningful model in predicting the catalytic properties. I am not an expert on the experimental aspects, thus my comments are mainly on the computational results. Many of the findings in this work are consistent with the general understanding of catalysts with nano-particles, thus are not surprising or completely new. For example, the findings that the interfacial atoms are more ionic and the oxidation state of Pt increases with the particle size decreases are such cases. However, they provide clear evidence that the CO oxidation activity is correlated with the ratio $N_{\text{neutral}}/N_{\text{cation}}/N_{\text{total}}$, which is interesting and provides insight for understanding the role of various metal sites with different oxidation states or charge states. I therefore think the manuscript might become publishable for Nature Communications after taking into account of the following comments and suggestions:

Response: We are grateful that the reviewer understands the importance of our work. Thanks to the reviewer's comments, the clarity of our work has been improved, as explained in the following point-by-point responses.

1) When Pt_n clusters are supported on oxides, the difference of the chemical potentials decides how much amount of charges are transferred, as have been discussed in the literature (e.g. J. Am. Chem. Soc., 2017, 139, 6190). They need to clarify this point because when the metal cluster and the support have similar chemical potentials, there is no cationic atoms, even at the interfacial region.

Response: We thank the reviewer for this comment regarding the charge transfer between the clusters and the support. Vila et al. have demonstrated charge transfer from Pt to an Al₂O₃ support, providing evidence for the existence of cationic atoms as shown in below¹.

FIG. 1. (Color) Time-elapsing rendering of the structure of a Pt₁₀ cluster on [110] γ -Al₂O₃. The purple and gold spheres represent Pt atoms that are metallic and oxidized, respectively, with the latter being bound to surface O atoms, while the red and turquoise spheres represent oxygen and aluminum atoms, respectively. The “blurriness” of a given atom characterizes its range of motion.

As we did not discuss charge transfer in the previous manuscript, to improve clarity, we revised the manuscript as follows.

On line 182, ‘This suggestion is reasonable because the coordination number of adsorbates has been shown to affect the adsorption energy² and because charge transfer from Pt to Al₂O₃ has been demonstrated¹.’

2) The undercoordinated, interfacial Pt atoms are more positively charged if the charge transfer flows from metal to support. Since the authors performed DFT calculations, I am surprised that they did not provide the Bader charges for these atoms at different layers of the cluster. The charges should be included to provide more direct data on the “chemical state” change of the Pt atom.

Response: In the previous manuscript, the results of charge calculations were not provided because Vila et al. have demonstrated charge transfer from Pt to an Al₂O₃ support.

However, in accordance with the reviewer’s suggestion, we performed DFT to provide the Bader charges for the Pt atoms and confirmed the occurrence of charge transfer from Pt to Al₂O₃ and cationic Pt atoms as shown in Supplementary Fig. 12. The manuscript was revised as follows.

On line 91 of the supplementary information, ‘Vila et al. have demonstrated charge transfer from Pt to an Al₂O₃ support¹. We have also achieved similar results, as

summarized in Supplementary Fig. 12d.'

Supplementary Fig. 12 | Average bond energy of Pt–Al₂O₃ and charge transfer from Pt to Al₂O₃. **a**, Pt/Al₂O₃ configurations used to obtain the binding energies of Pt clusters via DFT calculations. The solid rectangle (green) represents the unit cell, while the blue, red, and grey spheres represent Al, O, and Pt atoms, respectively. The integral numbers on the Pt atoms represent the Pt–Pt coordination numbers, which were used to calculate the average coordination number of the Pt atoms in the unit cell. **b**, Calculated

binding energy of Pt clusters as a function of the number of Pt atoms in the unit cell. **c**, Calculated average bond energy of Pt–Al₂O₃ as a function of the number of Pt atoms in the unit cell, in which the average bond energy was calculated by dividing the **binding** energy of the Pt cluster by the number of Pt atoms in the unit cell. **d**, **Structural models of Pt₁₀ cluster adsorbed on the Al₂O₃ surface constructed based on the ref. 1 to calculate Bader charges of the Pt atoms, demonstrating charge transfer from Pt to Al₂O₃ and existence of oxidized or cationic Pt atoms.**

3) Conceptually, valence state, oxidation state, and atomic charge are quite different concepts. The authors should avoid mixing them together. The high oxidation state metal (e.g. Pt(II) or Pt⁺²) often carries more positive charge than a low oxidation state metal (e.g. Pt(0)). But one cannot confuse oxidation state with charge state. In p. 7, “Pt within the Pt₇ cluster has an oxidation state between Pt₀ and Pt₂₊”. Here Pt₂₊ is the charge state, while Pt(II) or Pt⁺² is the oxidation state.

Response: We thank the reviewer for this comment on the use of the terms oxidation state and charge state. We revised all the relevant descriptions in the manuscript accordingly.

4) The key finding of CO oxidation activity to correlate with $N_{\text{neutral}}/N_{\text{cation}}/N_{\text{total}}$ needs some further explanation of its physical meaning. In fact, $N_{\text{neutral}}/N_{\text{cation}}/N_{\text{total}} = N_{\text{neutral}}/(N_{\text{cation}}N_{\text{total}}) = (N_{\text{total}} - N_{\text{cation}})/(N_{\text{cation}}N_{\text{total}}) = 1/N_{\text{cation}} + 1/N_{\text{total}}$; Because the N_{total} is usually far larger than N_{cation} , one can imagine that this simply says that the CO oxidation activity is inversely proportional to the N_{cation} .

Response: In this study, we obtained experimental evidence for the existence of cationic Pt atoms on an Al₂O₃ surface and we found that N_r/N_c is correlated with the CO oxidation activity. We also found that N_r/N_c divided by the number of atoms in the cluster shows a stronger correlation to the CO oxidation activity; however, the physical meaning of this parameter (division by n,) was unclear in the previous manuscript.

Through our attempts to find the physical meaning of this parameter (such as by transforming the equation as the reviewer also kindly attempted), we noticed that N_r/N_c is the atomic ratio of the neutral and cationic Pt atoms exposed on the

Pt_n/Al₂O₃ sample surface. Thus, N_n/N_c should be simply compared with the amount of CO₂ produced from that Pt_n/Al₂O₃ surface according to the different cluster sizes.

We deposited a certain amount of Pt atoms (~0.02 ML for all the Pt_n/Al₂O₃ samples) on the Al₂O₃ surface, and some of them appeared on the surfaces of clusters, which we defined as surface-exposed Pt atoms. It is logical to think that the CO₂ molecules detected by TPR were produced on these surface-exposed Pt atoms. Thus, if there are differences in the oxygen affinities of the neutral and cationic atoms, there should be some correlation between the atomic ratio of these sites and the amount of produced CO₂. For this reason, we think N_n/N_c should be compared with the amount of CO₂ produced for different *n* values of Pt_n/Al₂O₃ (as shown in Fig. 3g in the revised manuscript). An excellent R² value of 0.99 was achieved for the curves of N_n/N_c and the amount of CO₂ produced.

This consideration should also be applied to predict the ideal cluster size, and thus, Figs. 5b and 5c were also amended to compare the amounts of CO₂ produced with the ratios of neutral to cationic Pt atoms theoretically predicted using our BAM. These plots also exhibited higher R² values than those in the previous manuscript.

Figures 3 and 5 and the corresponding figure captions were revised as follows.

Fig. 3 | Chemical states of Pt and their effect on CO oxidation activity. a, IRAS spectra of ¹³CO adsorbed on Pt_n/Al₂O₃. **b**, Fraction of neutral and cationic Pt atoms estimated from IRAS spectra. **c**, Schematic diagram illustrating the adsorption of CO on a Pt cluster. **d**, Average bond energy of Pt–Al₂O₃ determined using DFT calculations. A linear increase in the bond energy between Pt and Al₂O₃ is observed as CN_{Pt–Pt} is decreased from 5 to 0. The structural models used for these calculations are shown in **Supplementary Fig. 12**. **e**, Schematic diagram of clusters showing the location of cationic Pt atoms for different Pt coordination numbers. **f**, Fraction of cationic Pt atoms in the cluster. Good agreement with the experimental data is observed for $X = 5$. **g**, Relationship between **the amount of produced CO₂ for Pt_n/Al₂O₃** and the ratio of neutral to cationic Pt atoms. The coefficient of determination (R²) between **these parameters** was calculated to be **0.99** using the least-mean square method.

Fig. 5 | Protocol for determining the optimum Pt cluster size for maximum CO oxidation activity, and comparison of CO oxidation activity and predicted atomic ratio. a, Schematic diagram illustrating the protocol used for optimum Pt cluster size prediction. The process starts with a theoretical calculation of the **binding energy** for a Pt monomer on the support material. **b,c**, Comparison of the atomic ratios of neutral to cationic Pt atoms with **the amounts of CO₂ produced** for Pt_n/Al₂O₃ (**b**) and Pt_n/TiO₂ (**c**).

On the other hand, when making this correlation of CO oxidation activity correlating with $N_{neutral}/N_{cation}/N_{total}$, have they excluded those atoms that are inaccessible (e.g. not on the surface) for CO oxidation?

Response: N_{total} is the total number of Pt atoms, including inaccessible atoms within the cluster in the previous manuscript, and thus, it was not appropriate to consider these atoms for surface reactions.

This comment is related to the previous comment given by this reviewer and as we stated above, by simply comparing $N_{neutral}/N_{cation}$ with the amount of produced CO₂ for Pt_n/Al₂O₃ (and for Pt_n/TiO₂), we achieved stronger correlations. We hope the reviewer finds our considerations and analyses technically sound.

5) The manuscript correlates the cationic state with the coordination number. In fact, both the coordination number and the bond distances

matter for the bond energy, as discussed in Nature Chem. 2015, 7, 403. The authors missed this important work.

Response: In the light of the reviewer's advice, we have cited the suggested article in the revised manuscript, as follows.

On line 181, 'This suggestion is reasonable because the coordination number of adsorbates has been shown to affect the adsorption energy² and because charge transfer from Pt to Al₂O₃ has been demonstrated¹.'

6) One should realize that the metal clusters might be dynamically changed in structure when CO and O2 are adsorbed and react (e.g. Nature Commun., 2015, 6, 6511). The intrinsic structure of Ptn/support may not be the same as it started before the reaction. This should be pointed out in the manuscript to readers.

Response: We agree with the reviewer about the possibility of dynamic changes to the cluster morphology during the catalysis process. We already mentioned such changes in Supplementary Note 1 in the previous manuscript, but to improve clarity, we added the following sentence in the revised manuscript, which cites the suggested literature.

On line 327, 'Furthermore, the cluster morphology might change dynamically during the reaction, as observed in several recent studies^{3,4}.'

7) Well-defined clusters embedded or supported on oxides surface provide opportunity for precision control of catalytic reactions. This kind of "single-cluster catalysts" are widely studied lately and should be discussed in the introduction or as perspective.

Response: In accordance with the reviewer's comment, we added the following text in the revised manuscript.

On line 328, 'Nonetheless, we believe that the results presented in this study will be helpful for designing atomically dispersed nanostructured catalysts including single-atom and few-atom clusters or single-cluster catalysts, which have recently received considerable interest^{5,6,7,8,9},

8) There are some typos, for instance, p. 9 "high electron negativity" should be "high electronegativity"; Pt-support interaction energy should be called "binding energy" not "adsorption energy". The latter is reserved for molecules (e.g. CO, O2, CO2) that adsorb and desorb on the

Ptn/support surface.

Response: We thank this reviewer for pointing out these typos. All the mentioned terms were revised accordingly (these changes are highlighted in red in the revised manuscript).

Reviewer #2 (Remarks to the Author):

This is a high-quality manuscript on the CO Oxidation reaction, where the authors try to provide a structure-activity relationship. Such knowledge is important and may be useful for purposeful catalyst design. Nevertheless, there are several issues requiring clarification.

Response: We thank the reviewer for understanding the importance of this work and are appreciative of the comments and remarks that have contributed towards improving our work. In the following, we provide point-by-point responses to the comments and address the issues raised by the reviewer.

i) A volcano-type dependence of catalyst activity on the number of Pt atoms in different clusters was established (Fig. 1b, Fig.3 g,h and Fig. 5 b,c). It is, however, not correct to relate the activity to the total number of Pt sites for the catalysts with 3-D structured Pt species.

In addition, it is not clear why the ratio of neutral to cationic Pt sites should be related to the total number of Pt atoms (See the right Y axis in Fig.3 h and Fig. 5 b,c).

Response: We thank the reviewer for this comment. We agree with the reviewer that it is not correct to relate the activity to the total number of Pt atoms because there are inaccessible atoms in the 3D structured Pt clusters.

Thanks to this comment, we noticed that that N_n/N_c is the atomic ratio of the neutral and cationic Pt atoms exposed on the Pt_n/Al_2O_3 sample surface. Thus, N_n/N_c should be simply compared with the amount of CO_2 produced from that Pt_n/Al_2O_3 surface according to the different cluster sizes.

We deposited a certain amount of Pt atoms (~ 0.02 ML for all the Pt_n/Al_2O_3 samples) on the Al_2O_3 surface, and some of them appeared on the surfaces of clusters, which we defined as surface-exposed Pt atoms. It is logical to think that the CO_2 molecules detected by TPR were produced on these surface-exposed Pt atoms. Thus, if there are differences in the oxygen affinities of the neutral and cationic atoms, there should be some correlation between the atomic ratio of these sites and the amount of produced CO_2 . For this reason, we think N_n/N_c should be compared with the amount of CO_2 produced for different n values of Pt_n/Al_2O_3 (as shown in Fig. 3g in the revised manuscript). An excellent R^2 value of 0.99 was achieved for the curves of N_n/N_c and the amount of CO_2 produced.

This consideration should also be applied to predict the ideal cluster size, and thus, Figs. 5b and 5c were also amended to compare the theoretically predicted amount of CO₂ produced using our BAM and experimental one. These plots also exhibited higher R² values than those in the previous manuscript.

Figures 3 and 5 and the corresponding figure captions were revised as follows.

Fig. 3 | Chemical states of Pt and their effect on CO oxidation activity. **a**, IRAS spectra of ^{13}CO adsorbed on $\text{Pt}_n/\text{Al}_2\text{O}_3$. **b**, Fraction of neutral and cationic Pt atoms estimated from IRAS spectra. **c**, Schematic diagram illustrating the adsorption of CO on a Pt cluster. **d**, Average bond energy of Pt– Al_2O_3 determined using DFT calculations. A linear increase in the bond energy between Pt and Al_2O_3 is observed as $\text{CN}_{\text{Pt-Pt}}$ is decreased from 5 to 0. The structural models used for these calculations are shown in **Supplementary Fig. 12**. **e**, Schematic diagram of clusters showing the location of cationic Pt atoms for different Pt coordination numbers. **f**, Fraction of cationic Pt atoms in the cluster. Good agreement with the experimental data is observed for $X = 5$. **g**, Relationship between **the amount of produced CO_2 for $\text{Pt}_n/\text{Al}_2\text{O}_3$** and the ratio of neutral to cationic Pt atoms. The coefficient of determination (R^2) between **these parameters** was calculated to be **0.99** using the least-mean square method.

Fig. 5 | Protocol for determining the optimum Pt cluster size for maximum CO oxidation activity, and comparison of CO oxidation activity and predicted atomic ratio. a, Schematic diagram illustrating the protocol used for optimum Pt cluster size prediction. The process starts with a theoretical calculation of the **binding** energy for a Pt monomer on the support material. **b,c**, Comparison of the atomic ratios of neutral to cationic Pt atoms with **the amounts of CO₂ produced** for Pt_n/Al₂O₃ (**b**) and Pt_n/TiO₂ (**c**).

ii) To support their conclusions, the authors should report surface coverage by adsorbed carbon monoxide and oxygen species before starting the reaction.

Response: In accordance with the reviewer's comment, the coverages of both CO and O are explicitly reported in the revised supplementary information. We did not detect O₂ desorption during the TPR measurements, as in a previous report¹⁰. Thus, based on this previous report, the O coverage was calculated by assuming that all the adsorbed O was consumed by CO during the reaction¹⁰. To report the coverages, we revised the manuscript as follows.

On line 112, 'The conversion efficiency from adsorbed CO to CO₂ and the **total amount of adsorbed CO and O for each cluster size are summarized in Fig. 1f and Supplementary Figs. 3a,b, respectively.**'

Supplementary Fig. 3 | CO and O coverages on mass-selected Pt_n clusters on Al_2O_3 . **a**, CO coverage. **b**, O coverage. The Pt_n deposited surfaces were exposed to 1000 L of $^{18}O_2$ at 300 K to saturate the Pt_n clusters with ^{18}O atoms, followed by saturation adsorption of ^{13}CO at 88 K, and finally, the TPR measurement was performed. The Pt coverage was 0.02 ML (1 ML = 1.5×10^{15} atoms/cm²). The CO coverage (**a**) was estimated as the sum of the amounts of produced CO_2 (Fig. 1b) and unreacted CO (Fig. 1d). We detected no O_2 desorption during the TPR measurement, as in a previous report. Thus, based on the previous report, the O coverage (**b**) was calculated by assuming that all the adsorbed O was consumed by CO during the reaction¹⁰. **c**, Ratio of CO coverage to O coverage. This ratio correlates well with $(N_n + N_c)/N_n$, suggesting that cationic Pt has a low oxygen affinity. $(N_n + N_c)/N_n$ was calculated from the IRAS results (Fig. 3).

Can the authors exclude the fact that the ratio of adsorbed carbon monoxide to oxygen species depend on the size of Pt clusters?

Response: If we understand the question correctly, then no, we do not think we can exclude the possibility of such a size dependence for the ratio of CO to O, as this trend is what we actually found in this study. To report the ratio of CO to O, we revised the manuscript as follows.

On line 221, 'In fact, θ_{CO}/θ_O was matched well with $(N_n + N_c)/N_n$ as shown in **Supplementary Fig. 3c.**'

Supplementary Fig. 3 | CO and O coverages on mass-selected Pt_n clusters on Al_2O_3 . **a**, CO coverage. **b**, O coverage. The Pt_n deposited surfaces were exposed to 1000 L of $^{18}O_2$ at 300 K to saturate the Pt_n clusters with ^{18}O atoms, followed by saturation adsorption of ^{13}CO at 88 K, and finally, the TPR measurement was performed. The Pt coverage was 0.02 ML (1 ML = 1.5×10^{15} atoms/cm²). The CO coverage (**a**) was estimated as the sum of the amounts of produced CO_2 (Fig. 1b) and unreacted CO (Fig. 1d). We detected no O_2 desorption during the TPR measurement, as in a previous report. Thus, based on the previous report, the O coverage (**b**) was calculated by assuming that all the adsorbed O was consumed by CO during the reaction¹⁰. **c**, Ratio of CO coverage to O coverage. This ratio correlates well with $(N_n + N_c)/N_n$, suggesting that cationic Pt has a low oxygen affinity. $(N_n + N_c)/N_n$ was calculated from the IRAS results (Fig. 3).

What is about the kind of adsorbed oxygen species?

Does oxygen exist in molecular or atomic forms or their mixtures?

Response: We think that the adsorbed oxygen species is atomic O. It is well known that adsorbed molecular O_2 dissociate to atomic O at ~ 150 K on Pt_n clusters¹¹ and at ~ 210 K on single crystalline Pt surfaces¹². As the adsorption temperature of oxygen in our experiments (300 K) is sufficiently higher than these dissociation temperatures, it is reasonable to consider that oxygen is adsorbed on Pt_n in the atomic form.

iii) The authors reported desorption profiles of CO in Fig. 1d. What is about oxygen?

Response: We detected no O₂ desorption during the TPR measurements, as also observed in a previous report¹⁰. This result suggests that all the O atoms reacted with CO (note, we detected unreacted CO but not O). Such discussion is now added in the revised manuscript as follows.

On line 418, 'We detected no O₂ desorption during the TPR measurements, as in a previous report¹⁰. Thus, based on the previous report, the O coverage was calculated by assuming that all the adsorbed O was consumed by CO during the reaction (Supplementary Fig. 3)¹⁰.'

iv) A weak point of the experimental part of this manuscript is that the authors do not provide the rate of CO oxidation.

Can the authors calculate such values from their data?

Response: The reaction rate is proportional to the activation energy (equation S1 below); thus, we estimated the reaction rate in the revised manuscript as follows.

On line 114, 'Interestingly, CO₂ is produced at the same temperature of ~300 K for all cluster sizes (Supplementary Note 1). This result indicates that the same activation barrier exists for each Pt cluster (Supplementary Note 2 and Supplementary Fig. 4).'

Supplementary Note 2.

Activation energies of CO₂ formation and CO desorption for Pt_n/Al₂O₃.

This study revealed structural sensitivity for a single catalytic reaction event, but the steady-state reaction rate (r) depends not only on the amount of adsorbate but also on the activation energy (E_a), as shown in equation (S1).

$$R = v \exp(-E_a/RT) \theta_{CO} \theta_O \quad (S1)$$

where v , R , and T are the pre-exponential factor, gas constant, and temperature, respectively. Therefore, from the slopes of the corresponding Arrhenius plots^{13,14}, we tentatively estimated the activation energies for CO₂ formation and CO desorption as 6~8 and 40~60 kJ/mol, respectively (Supplementary Fig. 4). Furthermore, these values were similar for the various Pt cluster sizes. Although our analyses might underestimate these values because the employed method does not consider the repulsive interactions between adsorbates^{15,16}, the value obtained for CO desorption is close to that for the

steady-state reaction (60.3 kJ/mol)¹⁷. The estimated activation energy of CO desorption is constant regardless of the cluster size, suggesting that the reaction rate under steady-state conditions would also show a size dependency similar to that of the single catalytic reaction event.

Supplementary Fig. 4 | Activation energies for CO₂ formation and CO desorption over Pt_n/Al₂O₃. **a**, CO₂ ($m/z = 47$) and CO ($m/z = 29$) TPR spectra over Pt₁₉/Al₂O₃. The Pt₁₉ deposited surface was exposed to 1000 L of ¹⁸O₂ at 300 K to saturate the Pt₁₉ clusters by ¹⁸O atoms, followed by saturation adsorption of ¹³CO at 88 K, and finally,

the TPR measurement was performed. **b**, Amounts of adsorbed ^{13}CO and ^{18}O over $\text{Pt}_{19}/\text{Al}_2\text{O}_3$ as a function of temperature. To estimate these amounts, the number of produced CO_2 molecules and unreacted CO molecules were determined from the TPR peak areas. The amount of adsorbed CO was determined by summing those of produced CO_2 and unreacted CO . The amount of adsorbed O was determined to be equal to that of produced CO_2 . **c**, Coverage-corrected Arrhenius plot for CO_2 formation over $\text{Pt}_{19}/\text{Al}_2\text{O}_3$. The activation energy was determined from the slope of the coverage-corrected Arrhenius plot^{13,14} (natural logarithm of $(r_{\text{CO}_2}/(\theta_{\text{CO}}\times\theta_{\text{O}}))$ versus reciprocal temperature, where r_{CO_2} represents the desorption rate of CO_2 , and θ_{CO} and θ_{O} represent the amounts of adsorbed CO and O , respectively). **d**, Size dependency of the activation energy for CO_2 formation. The error bar was determined from multiple measurements on a single cluster size. **e**, CO ($m/z = 29$) TPD spectrum over $\text{Pt}_{19}/\text{Al}_2\text{O}_3$. The Pt_{19} deposited surface was saturated by ^{13}CO at 88 K and then the TPD measurement was performed. **f**, Amount of adsorbed ^{13}CO over $\text{Pt}_{19}/\text{Al}_2\text{O}_3$ as a function of temperature. **g**, Coverage-corrected Arrhenius plot for CO desorption over $\text{Pt}_{19}/\text{Al}_2\text{O}_3$. The activation energy was determined from the slope of the coverage-corrected Arrhenius plot (natural logarithm of $(r_{\text{CO}}/\theta_{\text{CO}})$ versus reciprocal temperature, where r_{CO} and θ_{CO} represent the desorption rate of CO and the amount of adsorbed CO , respectively). **h**, Size dependency of the activation energy for CO desorption. **i,j**, Representative potential energy diagrams for CO oxidation over an oxygen-saturated Pt_n cluster (**i**) and a CO -saturated Pt_n cluster (**j**).

We hope the reviewer finds our considerations and analyses technically sound.

v) Will the authors observe the same activity-size dependence under steady-state conditions?

Response: If sufficient amounts of both CO and O_2 are provided, the reaction will depend on the atomic ratio or the oxygen affinity of the Pt atoms. Thus, based on this study, we think that we should see similar trend under steady-state conditions.

References

1. Vila F., Rehr J.J., Kas J., Nuzzo R.G. & Frenkel A.I. Dynamic structure in supported Pt nanoclusters: Real-time density functional theory and x-ray spectroscopy simulations. *Phys. Rev. B* **78**, 121404 (2008).

2. Calle-Vallejo F., Loffreda D., Koper Marc T.M. & Sautet P. Introducing structural sensitivity into adsorption–energy scaling relations by means of coordination numbers. *Nat Chem* **7**, 403-410 (2015).
3. Häkkinen H., Abbet S., Sanchez A., Heiz U. & Landman U. Structural, Electronic, and Impurity-Doping Effects in Nanoscale Chemistry: Supported Gold Nanoclusters. *Angew. Chem.-Int. Edit.* **42**, 1297-1300 (2003).
4. Wang Y.-G., Mei D., Glezakou V.-A., Li J. & Rousseau R. Dynamic formation of single-atom catalytic active sites on ceria-supported gold nanoparticles. *Nat. Commun.* **6**, 6511 (2015).
5. Beniya A. & Higashi S. Towards dense single-atom catalysts for future automotive applications. *Nat Catal.* **2**, 590-602 (2019).
6. Pérez-Ramírez J. & López N. Strategies to break linear scaling relationships. *Nature Catalysis* **2**, 971-976 (2019).
7. Khorshidi A., Violet J., Hashemi J. & Peterson A.A. How strain can break the scaling relations of catalysis. *Nature Catalysis* **1**, 263-268 (2018).
8. Liu J.-C., Ma X.-L., Li Y., Wang Y.-G., Xiao H. & Li J. Heterogeneous Fe₃ single-cluster catalyst for ammonia synthesis via an associative mechanism. *Nat. Commun.* **9**, 1610 (2018).
9. Xing D.-H., Xu C.-Q., Wang Y.-G. & Li J. Heterogeneous Single-Cluster Catalysts for Selective Semihydrogenation of Acetylene with Graphdiyne-Supported Triatomic Clusters. *J. Phys. Chem. C* **123**, 10494-10500 (2019).
10. Kaden W.E., Kunkel W.A., Kane M.D., Roberts F.S. & Anderson S.L. Size-Dependent Oxygen Activation Efficiency over Pd_n/TiO₂(110) for the CO Oxidation Reaction. *J. Am. Chem. Soc.* **132**, 13097-13099 (2010).
11. Heiz U., Sanchez A., Abbet S. & Schneider W.D. Catalytic Oxidation of Carbon Monoxide on Monodispersed Platinum Clusters: Each Atom Counts. *J. Am. Chem.*

Soc. **121**, 3214-3217 (1999).

12. Winkler A., Guo X., Siddiqui H.R., Hagans P.L.&Yates J.T. Kinetics and energetics of oxygen adsorption on Pt(111) and Pt(112)- A comparison of flat and stepped surfaces. *Surface Science* **201**, 419-443 (1988).
13. Holmes Parker D., Jones M.E.&Koel B.E. Determination of the reaction order and activation energy for desorption kinetics using TPD spectra: Application to D2 desorption from Ag(111). *Surf. Sci.* **233**, 65-73 (1990).
14. Hopstaken M.J.P.&Niemantsverdriet J.W. Structure sensitivity in the CO oxidation on rhodium: Effect of adsorbate coverages on oxidation kinetics on Rh(100) and Rh(111). *J. Chem. Phys.* **113**, 5457-5465 (2000).
15. Miller J.B., *et al.* Extraction of kinetic parameters in temperature programmed desorption: A comparison of methods. *J. Chem. Phys.* **87**, 6725-6732 (1987).
16. Nieskens D.L.S., van Bavel A.P.&Niemantsverdriet J.W. The analysis of temperature programmed desorption experiments of systems with lateral interactions; implications of the compensation effect. *Surf. Sci.* **546**, 159-169 (2003).
17. Yin C., *et al.* Alumina-supported sub-nanometer Pt₁₀ clusters: amorphization and role of the support material in a highly active CO oxidation catalyst. *J. Mater. Chem. A* **5**, 4923-4931 (2017).

REVIEWERS' COMMENTS:

Reviewer #1 (Remarks to the Author):

The authors have revised the manuscript thoroughly and addressed my concerns well. They also provide further justification of the correlation of activity and the N_n/N_c ratio, which I think provides insights for future design of single-atom catalysts and single-cluster catalysts. I therefore believe the current version of manuscript is ready for publication on Nature Communications.

Reviewer #2 (Remarks to the Author):

The authors have clarified my previous comments to a large extent and modified the manuscript and the Supplementary Information. I have only minor comments related to the new results shown in Supplementary Fig. 4. What do desorption rates of CO and CO₂ really mean? The authors should write how the rates were determined experimentally. Why do the Arrhenius plots in (b) and (f) have such strange profiles at high temperatures?

Responses to the comments from reviewers

CO oxidation activity of non-reducible oxide-supported mass-selected few-atom Pt nanoclusters

Atsushi Beniya^{1,*}, Shougo Higashi^{1,*}, Nobuko Ohba¹, Ryosuke Jinnouchi¹, Hirohito Hirata² & Yoshihide Watanabe¹

¹Toyota Central R&D Labs, Inc., 41-1 Yokomichi, Nagakute, Aichi 480-1192, Japan

²Toyota Motor Corporation, 1200 Mishuku, Susono, Shizuoka 410-1193, Japan

We thank the reviewers for their comments and insightful remarks. All the points made here are reflected in the revised manuscript. The reviewers' comments are shown in ***italic bold*** text. In our responses, the text shown in red indicates added or revised sentences.

Reviewer #1 (Remarks to the Author):

The authors have revised the manuscript thoroughly and addressed my concerns well. They also provide further justification of the correlation of activity and the Nn/Nc ratio, which I think provides insights for future design of single-atom catalysts and single-cluster catalysts. I therefore believe the current version of manuscript is ready for publication on Nature Communications.

Response: We appreciate the reviewer comments to improve the clarity of our work and constructive remarks especially for Nn/Nc ratio. We hope this work will be a help for future design of single-atom catalysts and single-cluster catalysts.

Reviewer #2 (Remarks to the Author):

The authors have clarified my previous comments to a large extent and modified the manuscript and the Supplementary Information.

I have only minor comments related to the new results shown in Supplementary Fig. 4. What do desorption rates of CO and CO₂ really mean? The authors should write how the rates were determined experimentally.

Response: We thank the reviewer for comments and remarks through the revisions. In accordance with the reviewer's comment, we added the following to describe how we determined the desorption rates of CO and CO₂ in the revised manuscript.

On line 423, "Desorption rates of CO and CO₂ from the sample exposed to ¹³CO and ¹⁸O₂ were estimated by measuring the corresponding ion currents using QMS, whereby the sample surface was heated at a constant rate (3.5 K s⁻¹) and ion current is continuously monitored."

The desorption rate of CO shown in Supplementary Fig. 4a represents the unreacted CO molecules rates at different temperatures from the sample which is exposed to ¹³CO and O₂. Desorption rate of CO₂ is also shown in the same figure. Our answer to the question from the reviewer is that these desorption rates results with respect to temperature mean that CO₂ is produced and desorbed from the surface followed by desorption of the unreacted residual CO from the surface.

Why do the Arrhenius plots in (b) and (f) have such strange profiles at high temperatures?

Response: Honestly, the reason is not clear for non-linear behaviour at high temperatures. As one of the possibilities, dynamical structural change of cluster might be occurred at this high temperature region, which induces an increase of the entropy of the cluster, resulting in a different pre-factor.